# Analyzing preventive precautions to limit spread of COVID-19

**Ayaz Ahmad[1], Furqan Rustam[2], Eysha Saad[1], Muhammad Abubakar Siddique[1], Ernesto Lee[3], Arturo Ortega Mansilla[4,5], Isabel de la Torre Díez[6]\*, Imran Ashraf[7]\***

**1** Department of Computer Science, Khawaja Fareed University of Engineering and Information Technology, Rahim Yar Khan, Pakistan, **2** Department of Software Engineering, School of Systems and Technology, University of Management and Technology Lahore, Lahore, Pakistan, **3** Department of Computer Science, Broward College, Broward County, Florida, United States of America, **4** European University of The Atlantic, Santander, Spain, **5** Iberoamerican International University, Campeche, Mexico, **6** Department of Signal Theory and Communications and Telematic Engineering, Unviersity of Valladolid, Valladolid, Spain, **7** Information and Communication Engineering, Yeungnam University, Gyeongsan, Korea

\* isator@tel.uva.es (ITD); imranashraf@ynu.ac.kr (IA)

**Data Availability Statement:** The collected data is publicly available at this link: (https://www.kaggle.com/datasets/furqanrustam118/covid19-precautions-related-tweets).

## Abstract

With the global spread of COVID-19, the governments advised the public for adopting safety precautions to limit its spread. The virus spreads from people, contaminated places, and nozzle droplets that necessitate strict precautionary measures. Consequently, different safety precautions have been implemented to fight COVID-19 such as wearing a facemask, restriction of social gatherings, keeping 6 feet distance, etc. Despite the warnings, highlighted need for such measures, and the increasing severity of the pandemic situation, the expected number of people adopting these precautions is low. This study aims at assessing and understanding the public perception of COVID-19 safety precautions, especially the use of facemask. A unified framework of sentiment lexicon with the proposed ensemble EB-DT is devised to analyze sentiments regarding safety precautions. Extensive experiments are performed with a large dataset collected from Twitter. In addition, the factors leading to a negative perception of safety precautions are analyzed by performing topic analysis using the Latent Dirichlet allocation algorithm. The experimental results reveal that 12% of the tweets correspond to negative sentiments towards facemask precaution mainly by its discomfort. Analysis of change in peoples' sentiment over time indicates a gradual increase in the positive sentiments regarding COVID-19 restrictions.

## Introduction

The outbreak of the COVID-19 pandemic has resulted in devastating effects worldwide with a huge human and financial loss. According to the World Health Organization (WHO), 2% population (168 million) of the world was infected by COVID -19 after one year of its detection in Wuhan China, and 3.5 million people out of 168 million lost their lives [1]. Report on the economy shows that the COVID-19 pandemic will cause extreme poverty for 96 million people by 2021and out of 97 million people 47 million are women [2]. The COVID-19 outbreak

**Funding:** This research was supported by the European University of The Atlantic. This research was supported by the Florida Center for Advanced Analytics and Data Science funded by Ernesto.Net (under the Algorithms for Good Grant). The funders had no role in study design, data collection and analysis, decision to publish, or preparation of the manuscript.

**Competing interests:** The authors have declared that no competing interests exist.

affected a large population around the globe and continues to inflict financial and human suffering as the number of infected people keeps on rising.

The severe acute respiratory syndrome coronavirus 2 (SARS-CoV-2) is causing the COVID-19 that mainly attacks the respiratory system [3]. Over time, infected people exhibit various symptoms like dry cough, mild to high temperature, chest pain, and distress, pneumonia, breathing difficulty, and cardiac complications [4]. The symptoms of COVID-19 may vary from person to person and the symptoms grow severe for infected people with preexisting medical conditions such as respiratory problems, diabetes, cardiac disorders, tuberculosis, etc. [5]. Also, the people facing COVID-19 are prone to the risk of other diseases, especially mental and psychiatric disorders, as reported in [6, 7]. Such influence is different regarding both the age and gender of the infected people [8, 9]. In addition, several factors increase the probability of death by COVID-19 like the higher age, preexisting medical conditions including diabetes, cardiac complications, and respiratory problems, and longer admission time [7, 10].

Keeping in view the surge in the infected people, health institutions and governments introduced corresponding safety precautions to curb the spread of the virus such as keeping a distance of 6 feet, sanitizing the area and persons, and wearing facemasks, etc. [11]. Such precautionary measures are imposed by the Governments to limit the spread of COVID-19 as they are the first defense against the pandemic. One of the most important measures is washing hands thoroughly and frequently for 30 seconds. Also, the use of hand sanitizer is implemented. Similarly, people are advised to avoid contact with the face, especially the nose to reduce the risk of infection [12]. Washing hands diligently also helps to reduce COVID-19 transmission. In addition, respiratory hygiene like covering the face while coughing is advised to curb the transmission. Avoiding crowds and public places, staying and working from home, and reducing close contact with other people are additional precautionary measures that can help in controlling the spread of COVID-19 [13]. Prevalent consequences are observed when these precautions are not embraced and followed by the public. Previous studies manifested that the spread of the virus can be minimized if the public follows the precautionary measures defined by the government and healthcare officials [14–16]. Although COVID-19 policies are designed and deployed for the safety and well-being of people; such policies often create disputes and dissent. A large number of people demonstrate an unwillingness to follow such precautions, especially wearing the facemask [17, 18]. People think that wearing a face mask is difficult and may lead to additional respiratory complications if used for a longer period. So, investigating the views of the masses regarding such restrictions is an important research area to revise policies to both resolves people's disagreements and protect them from the disease.

People share their experiences, opinions, and perceptions regarding COVID-19 restrictions on social media platforms. Twitter is a well-known microblogging platform where users share their thoughts, views, and opinions in different groups as short texts called 'tweets'. These tweets contain information regarding public opinion towards people, policies and procedures, etc. For deeper insights into public opinion, sentiment analysis can be performed on these tweets and their positivity and negativity can be determined [19]. In this regard, analysis of tweets regarding facemask restriction can be very helpful to determine people's perceptions and devise the corresponding policies to increase the public acceptance of such restrictions since sentiment analysis aims to filter out the positive, negative, and neutral sentiments from the data [20].

Twitter provides a large corpus of text data that can assist in characterizing the current and everyday happenings in real-time. As a result, it is useful for corporations and people who are interested in keeping tabs on public health and social concerns. Following the COVID-19 precautions fits under both categories since wearing the mask has been scientifically proven to contain the spread of COVID-19 along with keeping a distance of 6 feet and using a sanitizer

however these precautions have faced significant backlash by the public for a variety of socio-political agendas. Given the extent to which the public's perception of COVID-19 has evolved with time and the spread of COVID-19 into a global disaster, the research related to studying the disparity of following COVID-19 precautions has significantly increased [21]. Accustomed to the unusual nature of the virus pandemic, it is significant to analyze the current sentiment of the public towards COVID-19 precautionary measures such as facemasks, keeping 6 feet distance, and using hand sanitizers. Predominantly, existing studies follow machine learning-based approaches for analyzing the sentiments for several reasons. Datasets contain a large number of tweets, views, and short texts, and manual annotation and analysis of such a large dataset are laborious and time-consuming. Such issues can be resolved by using machine learning models. Human experts are subject to error, and bias due to mood, fatigue, and other factors which may lead to wrong annotation and sentiment classification. Also, the performance of machine learning can be enhanced by following parameter optimization, data cleaning, and appropriate feature selection. Given these advantages, and embracing the use of machine learning approaches by existing studies, this research adopts a machine-learning-based methodology. It follows the similar flow of data preprocessing, feature extraction, training, and evaluation of machine learning models as embraced by existing studies. This methodology eventually leads to two advantages in essence. First, the study follows the norms of the research community that has been well established. Secondly, the use of the same benchmark makes it easy to analyze its performance with respect to existing works. This study aims at the following objectives

1. Analyzing the sentiments of the public towards the use of face masks which is imposed to limit the spread of COVID-19.

2. Finding and discussing the problems people experience while wearing the face mask.

This study proposes a hybrid approach to perform sentiment analysis of tweets related to safety precautions for COVID-19 and makes the following contributions

- A unified framework of sentiment lexicon and machine learning classifier has been proposed where the proposed model is a voting classifier, extreme boosting-decision tree (EB-DT), and integrates extra tree classifier (ETC), gradient boosting machine (GBM), and DT models. For experiments, different sets of data are extracted from Twitter. Three different datasets are extracted, to be precise, for analyzing the change in peoples' sentiments over time.

- Machine learning models are also employed for performance comparison including Ada-Boost classifier (AC), ETC, logistic regression (LR), random forest (RF), GBM, K nearest neighbors (KNN), support vector classifier (SVC), and DT which are used with term frequency-inverse document frequency (TF-IDF) and bag of words (BoW). Experiments also involve long short-term memory (LSTM), convolutional neural network (CNN)-LSTM, and gated recurrent unit (GRU) with a large dataset collected from Twitter. The lexicon-based TextBlob is used to annotate the collected dataset.

- Topic modeling of positive and negative is carried out using the Latent Dirichlet allocation (LDA) algorithm to analyze the factors that negatively influence the safety precautions against COVID-19.

- The accuracy of the proposed framework regarding the sentiment classification is compared against a professional sentiment analysis app called 'sentiment viz app'.

The rest of this paper breaks down as follows. Important works related to the current study are described in the following section. It is followed by the discussion the proposed framework,

voting classifier, dataset collection, and other details. Results and discussions are given after that. In the end, the study is concluded.

## Related work

Since the emergence of COVID-19 the discussion on Twitter has predominantly shifted towards the pandemic. People share their perceptions regarding the events occurring during the pandemic and policies being made by the government to limit the spread of the virus which has attracted the majority of researchers to analyze the overall public perception of COVID-19 and its relevant policies. The use of social media platforms experienced a surge amid the COVID-19 pandemic. The study [22] suggests that the use of social media has increased by 61% since the pandemic. The shift in connectivity to the virtual platforms has been largely attributed to the dissemination of information regarding COVID-19. The study [23] states that a significant amount of information and news is shared via such platforms to spread awareness.

The study [23] utilized tweets of Indian users to explore the sentiments towards lockdown due to COVID-19. The analysis is performed on extracted 24,000 tweets for 3 days. The analysis reveals that despite the negative sentiments about the lockdown, a large portion of positive sentiments is also observed. Some tweets expressed trust towards the governmental policy regarding the lockdown while other tweets showed that people were surprised by the decision. Overall, a positive attitude is observed towards the lockdown precaution to limit the spread of COVID-19. The study [16] conducted an extensive sentiment analysis of tweets related to COVID-19 hashtags to assist the government officials and healthcare professionals in making the decision and changing policies according to the public perception using a variety of machine learning models. Another study [24] employed a publicly available sentiment lexicon to get an understanding of the public opinions towards the COVID-19 epidemic. The study revealed that the ratio of positive, negative, and neutral tweets was 36:14:50 showing that a small portion of people has a negative perception of the epidemic.

A statistical analysis of COVID-19 related tweets is conducted in study [25] indicating that on average only a few people wrote negatively regarding the COVID-19 epidemic, and a large portion of people showed positive perception towards the precautionary measures. Similarly, a study [26] specifically revealed the perception of Nepalis people towards the COVID-19 pandemic and implemented policies to reduce the spread of COVID-19. The study analyzed the tweets of Nepalis using TextBlob and showed that the people of Nepal have a positive and hopeful attitude towards the pandemic. Many studies utilized English tweets to analyze the public perception, however, [27] integrated Arabic tweets to understand public opinions towards the COVID-19 pandemic by using the emoji lexicon.

For analyzing the public perception of the COVID-19 pandemic and the implementation of strict policies around the globe, many researchers investigated the factors impacting the perception of people. Topic modeling techniques have been utilized to investigate the most frequent factors and topics that highly impact the overall sentiment of the public. For instance, the authors [28] extracted topics from COVID-19 related tweets using LDA and showed that an overall positive perspective of people is observed over 8 topics whereas, the sentiments are negative for only 2 topics corresponding to the fatality rate of COVID-19. In another topic analysis conducted by [29], the authors clustered the data from five different social media platforms including Gab, Reddit, YouTube, Instagram, and Twitter by utilizing cosine similarity integrated into PAM (Partitioning Around Medoids).

They identified the topics individually for each platform and revealed that 21 topics are found on Twitter including racism, symptoms, disease description, etc. Consequently,

another study [30] investigated 20 million tweets from around the world in the time frame of 3 months to understand the trend of people's perception using the CrystalFeel algorithm. The study revealed that the extracted topics are correlated with the fear regarding the shortage of medical supplies and COVID-19 tests along with anger towards the lockdown. Sadness is correlated with losing friends and family members however, joy involved the words such as good health and gratitude. Another study [31] annotated the Twitter dataset consisting of tweets of users related to COVID-19 into seventeen topics extracted by LDA. The study revealed that anger is the most significant topic. The proposed model is a hybrid model that shows superior results for sentiment classification. Soft voting criteria are used to combine three models, similar to several previous studies on sentiment prediction. The study [32] proposed an ensemble model for the medical review sentiment predictions. The authors used soft voting criteria to combine LR, RF, and DT and achieved significant results on medical data. The study [33] used voting criteria to combine the GBM and SVM for user reviewer sentiment classification. Soft voting criterion is used to combine the models which are trained using several features such as uni, bi, and tri grams. The hybrid GBSVM outperforms with a significant 90% accuracy in comparison to other models. Similarly [34] also used soft voting criteria to combine two models LR and stochastic gradient classifier for tweets sentiment classification. The proposed models LR-SGDC perform significantly better with the soft voting approach to achieve high accuracy.

On top of sentiment analysis works, several studies investigate the relationships between various environmental factors and COVID-19. For example, [35] proposed an approach to find the relationship between COVID-19-related deaths, PM10, economic growth, PM2.5, and NO2 concentrations in New York state. The authors used city-level data to analyze the pattern using machine learning models. Similarly, [36] adopted a machine learning approach to find the relationship between pollution, economic growth, and deaths in India. The influence of NO2 concentration on COVID-19 deaths is investigated in [37] within the context of France. An artificial neural network has been adopted with a causal direction from the dependency algorithm for the study. The study successfully determines the NO2 thresholds concerning the spread of COVID-19. Similarly, the influence of particulate matter on public health is investigated in [38] for French cities.

Despite the sentiment analysis of tweets regarding safety precautions against COVID-19, people's perceptions and conceptions of the facemask have not been extensively investigated. This study aims at finding the factors influencing people's mistrust of safety precautions.

## Materials and methods

This study aims at exploring the sentiments of people's opinions towards following COVID-19 safety precautions by leveraging the machine learning-based framework. The architecture of the proposed framework is shown in Fig 1.

The proposed framework consists of data collection, preprocessing, data annotation using TextBlob, feature extraction, and training and testing the models. Different sets of data are collected during different periods to analyze the change in peoples' sentiments regarding COVID-19 restrictions. Annotation is carried out using lexicon-based TextBlob to avoid the limitation of error-prone and biased labeling by human experts. Feature extraction involves TF-IDF and BoW, the two most commonly used feature engineering approaches for text classification and sentiment analysis. Preprocessing is done to increase the efficacy of the training process. In the end, testing is performed on the unseen data to investigate the efficiency of the machine learning models. These steps are briefly described in the following sections.

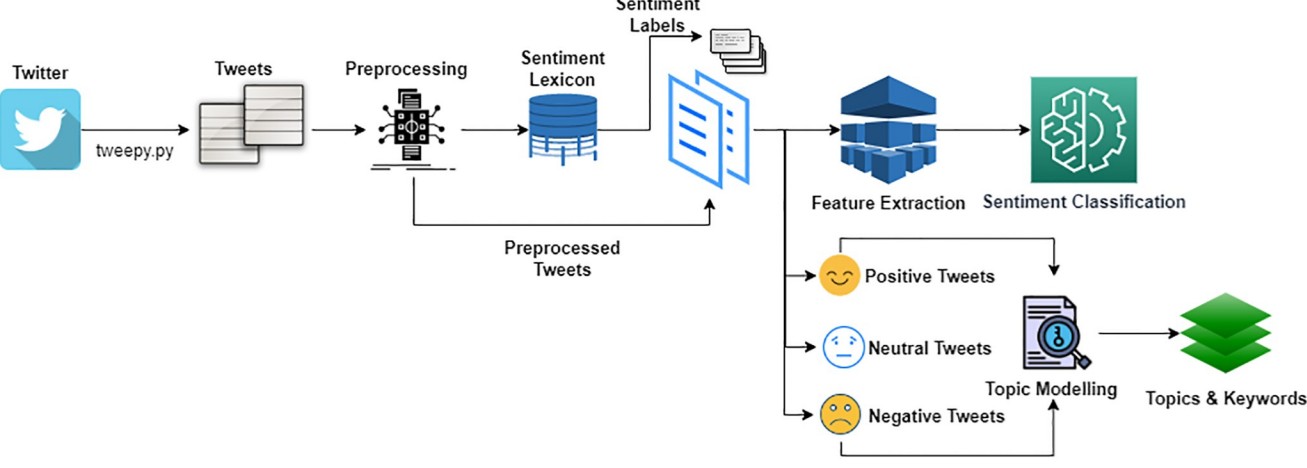

**Fig 1. General framework of the proposed sentiment classification process.**

## Dataset collection

For sentiment analysis, a dataset is extracted containing from Twitter. Twitter is used for data collection due to several factors. First, Facebook follows a friend-based network where only the people in the friend list can view the information on a profile page [39] which makes it private or closed. Hence, the data collection from Facebook to perform sentiment analysis is limited. Second, Twitter provides access to tweets with a 'developer' account and data collection is open and easy. Third, In addition, Twitter provides an application programming interface (API) that simplifies the task of data collection. Last and foremost is the length of tweets which is limited by 140 characters which makes it easy to perform analysis on short messages as compared to other social media platforms. A total of 8911 tweets from Twitter related to four hashtags including #facemask, #facemaskprecautions, #covid19, #covid19precautions by utilizing the Tweepy library. We collect this data during January 2022 during the peak period of the COVID-19 5th wave. The dataset contains three attributes including user ID, location, and tweet text. A few sample tweets from the dataset are shown in Table 1.

## Data preprocessing

The collected text is further subjected to preprocessing as it includes unnecessary data such as hashtags, URLs, punctuation marks, et. We utilized a variety of processes using the NLTK (Natural Language Toolkit) to clean the data [40]. Tokenization is one of the foremost tasks as it breaks down a text into small chunks known as tokens. In this study, we implemented RegexpTokenizer in Python language on the text data to provide the computer with a particular number of tokens that it can process to provide the desired outcome.

**Table 1. Text from sample tweets.**

| Location | Text |
|---|---|
| Los Angeles | good to be back at the gym today—face mask n all! #fitness #health #workout #exercise #gym #weightsâ€—https://t.co/dZagiNIZCV |
| Orange County, CA. | One of two shots so far, but have always, and will continue to wear the mask. #FaceMask https://t.co/18lmUwgW5h |

The tweets are the text which also includes punctuation marks such as "!@#$%&̂*()". The punctuation characters are not significant for the sentiment of the tweet and thus are regarded as unnecessary. Therefore, we removed them to clean the data for the learning models to interpret the text data with more efficacy. The learning models require data that is relevant for the analysis such as a tweet containing the number "4" or "3" does not have any impact on the sentiment of the tweet. Accordingly, null values are the noise that leads to ambiguous results. Therefore, numeric values are removed to enhance the performance of learning models.

The extracted tweets contain a variety of hashtags along with the URL which does not contribute to the sentiment of the text. Therefore, the hashtags and URLs are removed from the tweets which result in less noise and more relevant data for the learning models Stop-words are the most recurring words in the text document and do not contribute to the learning of machine learning models. Therefore, we imported the stopwords library from the NLTK and removed the stopwords from the tweets. Similarly, the learning models are case sensitive, such as they consider the words 'Good' and 'good' as two different words. It increases the feature space and the computational complexity of the model. For text classification, case normalization helps to improve the performance of learning models.

Stemming and lemmatization are two techniques that involve the transformation of a word into its root form. Stemming works by removing the beginning or end of an inflected word by taking prefixes and suffixes into account. However, removing the suffixes and prefixes might sometimes not be successful. Therefore, to overcome this limitation of stemming, lemmatization has also been integrated into this study. Lemmatization helps to transform the extended words into their base form and thus reduces the complexity of feature space by reducing the number of unique words as extended forms of words are considered different words from machine learning classifiers. Lemmatization differs from stemming in terms of considering the morphological analysis of the inflected words [41]. In this study, we utilized PorterStemmer and WordNetLemmatizer for stemming and lemmatization, respectively [41].

Table 2 shows the text from sample tweets before and after the above-mentioned preprocessing steps are carried out. It can be observed that numbers, punctuation, stopwords, URLs, and special symbols have been removed and the tweet text has been tokenized.

## TextBlob

For a supervised sentiment analysis of tweets, labeled data is required. TextBlob is used to annotate the tweets related to COVID-19 safety precautions. TextBlob is a Python library that is widely used to process data. It includes API (Application Programming Interface) to perform several NLP (Natural Language Processing) tasks like phrase extraction, translation and sentiment analysis, etc. [42]. We categorize the data into three classes positive, negative, and neutral. Among a total of 8911 records, TextBlob categorized 3193 records as positive, 4599 records as neutral while 1119 are labeled as negative, as shown in Fig 2.

**Table 2. Sample tweets text before and after preprocessing steps.**

| Original text | Preprocessed text |
|---|---|
| good to be back at the gym today—face mask n all! #fitness #health #workout #exercise #gym #weightsâ€—https://t.co/dZagiNIZCV | "good", "back", "gym", "today", "face", "mask" |
| One of two shots so far, but have always, and will continue to wear the mask. #FaceMask https://t.co/18lmUwgW5h | "one", "two", "shot", "always", "continue", "wear", "mask" |

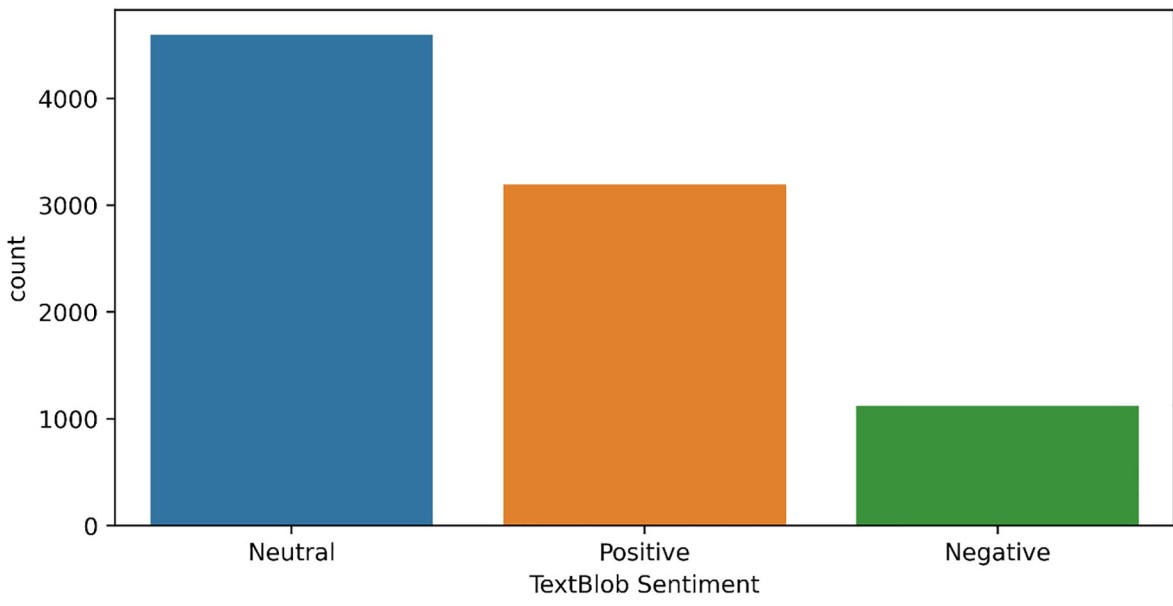

**Fig 2. Number of samples for neutral, positive and negative sentiments by TextBlob.**

Fig 3 shows the ratios of positive, negative, and neutral sentiments, as labeled by TextBlob. It can be observed that neutral sentiments make up most of the dataset with 51.61% of the dataset. It is followed by the positive and negative tweets which make 35.83% and 12.56%, respectively.

### Feature extraction

Learning algorithms are highly dependent on the features to make accurate predictions [43]. This makes extraction of features a very crucial step as highly correlated features will produce a better outcome and highly accurate classification and vice versa. Often, it is better to utilize more than one feature extraction approach to analyze their efficacy. This study follows a similar norm and selects TF-IDF and BoW as feature extraction techniques.

The BoW is a flexible and simple feature extraction technique that is defined as the representation of text data within a document. It quantifies the frequency of the occurrence of the word in the document irrespective of its position and importance in the document [44]. BoW mainly involves two metrics including the vocabulary of words that are known by the model and the measure of the occurrence of those particular words which are present in the vocabulary.

TF-IDF is a statistical technique that assesses the relevance of a word in a document or a collection of documents. TF-IDF not only vectorizes the text data but also quantifies the features among the whole corpus [45]. TF-IDF is obtained by taking a product of two metrics including the number of times a word has appeared in a document and the IDF (inverse document frequency) of a word in a collection of documents [46]. It assigns weights with respect to the importance of a term in a given corpus.

### Machine learning models

Sentiment classification models are the learning algorithms that categorize the given text into three sentiments. Several well-known machine learning models have been selected for the task

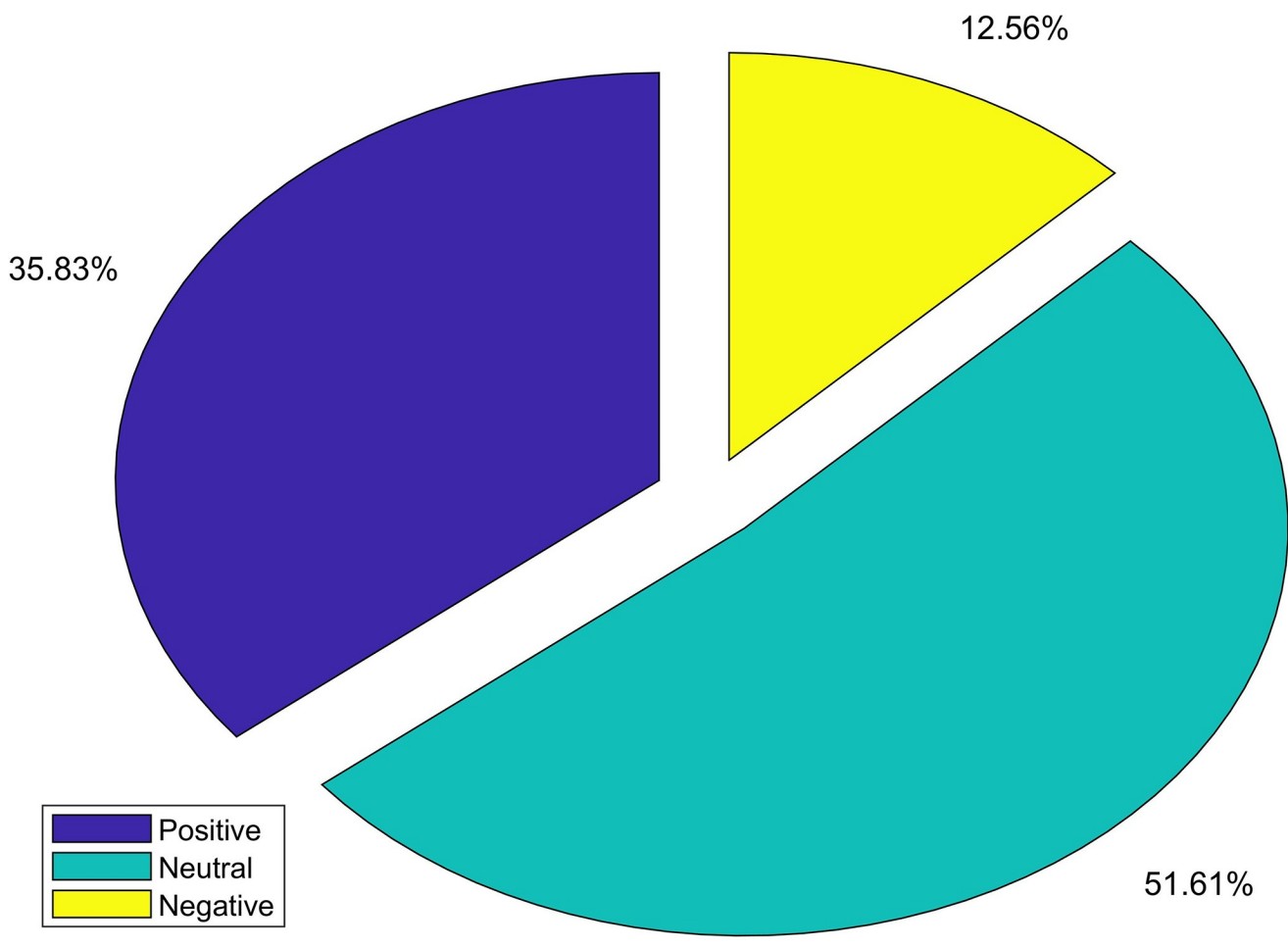

12.56%

35.83%

51.61%

**Positive**
**Neutral**
**Negative**

**Fig 3. Ratio of positive, negative and neutral sentiments.**

at hand in this study including AC, ETC, LR, RF, GBM, KNN, SVC, and DT. These models are optimized regarding their prediction accuracy by using different sets of hyperparameters where a complete list of hyperparameters used in this study is provided in Table 3.

**Adaptive boosting classifier.** AC also known as adaptive boosting is an ensemble machine learning classifier that iteratively constructs the ensemble model by combining multiple classifiers into a strong classifier. Fundamentally, it involves one-level DTs as weak learners

**Table 3. Hyperparameter settings of machine learning sentiment classifiers.**

| Classifier | Hyperparameter settings |
|---|---|
| RF | n_estimators = 600, random_state = 150, max_depth = 400 |
| LR | random_state = 50, solver='saga', multi_class='multinomial', C = 3.0 |
| ETC | n_estimators = 100, random_state = 150, max_depth = 400 |
| GBM | n_estimators = 100, learning_rate = 1, random_state = 50 |
| KNN | n_neighbors = 7 |
| SVC | kernel='linear', C = 1.0, random_state = 500 |
| AC | n_estimators = 100, random_state = 50 |
| DT | random_state = 150, max_depth = 400 |

which are then added sequentially to the ensemble model [47]. It follows two conditions: the training of the classifier should be interactive on a variety of training samples that are given specific weights, and training error is minimized with each subsequent iteration.

**Extra tree classifier.** ETC is a supervised ensemble classifier that aggregates the output of multiple DTs which are de-correlated to provide the final prediction [48]. ETC is similar to RF but differs in the manner in which DTs are constructed in the forest. The original training sample is integrated to train each DT in the ensemble of ETC resulting in a reduction of bias. Furthermore, each DT is provided with the k-features from the random set of sample records from which the best features are selected by the DTs to split the node based on criteria like the Gini Index. The random sample of k-features results in the construction of multiple de-correlated DTs.

**Logistic regression.** LR is a statistical supervised machine learning algorithm that performs classification based on probability by using the logistic function also known as the Sigmoid function [49]. The probability ratio in logistic regression is directly modeled. For predicting the probabilities, the Sigmoid function limits the real value into the range of 0 and 1 by making an S-shaped curve.

**Random forest.** RF constructs an ensemble of DTs hence named the 'Forest'. The DTs in RF are primarily trained using the bootstrap aggregation method [50] which is also referred to as 'Bagging' as it trains an individual DT in the ensemble with a random sample of the training set. After generating several data samples, the weak learners such as individual DTs in the forest of RF are trained independently. RF provides more randomness to the DTs while they are trained. Rather than selecting the significant feature while the node is split, it explores the optimum features from the random set of the features resulting in a more diverse nature of the model.

**Gradient boosting machine.** GBM is a boosting algorithm that reasons that the subsequent model when merged with the preceding model decreases the overall error rate in data classification [51]. Mainly, a target is set for the subsequent model to decrease the prediction error. The target for each subsequent model depends on the change in the prediction error. The target of each weak learner is set based on the error gradient corresponding to the prediction. Each subsequent model attempts to minimize the error in the prediction in the training phase [52].

**K nearest neighbor.** KNN is the supervised machine learning algorithm that is used for both classification and regression tasks [53]. It is easy to interpret, involves less calculation time, and has more predictive power. It works by assuming the correspondence between the trained data instances and the test or unseen data instances. It does not make any assumptions regarding the underlying data thus making it a non-parametric model.

**Support vector classifier.** SVC works based on pattern analysis and is implemented by integrating the linear kernel functions which enable the algorithm to map the input textual feature set into a vector space of higher dimensionality [16]. Afterward, the margin between the input features and the target variable is maximized to construct a linear hyper-plane. The classification task is accomplished by finding a hyper-plane that allows two classes to be finely separated.

**Decision tree.** DT is a well-known supervised machine learning algorithm that is widely used for classification tasks [54]. DT has a tree-like structure where the prediction process is carried out from node to leaf. DT is constructed and trained on the training set recursively. The learning phase terminates when no more splitting is possible or the output at the node is the same as the target label or class. DT is good for the discovery of useful and meaningful patterns from the input data.

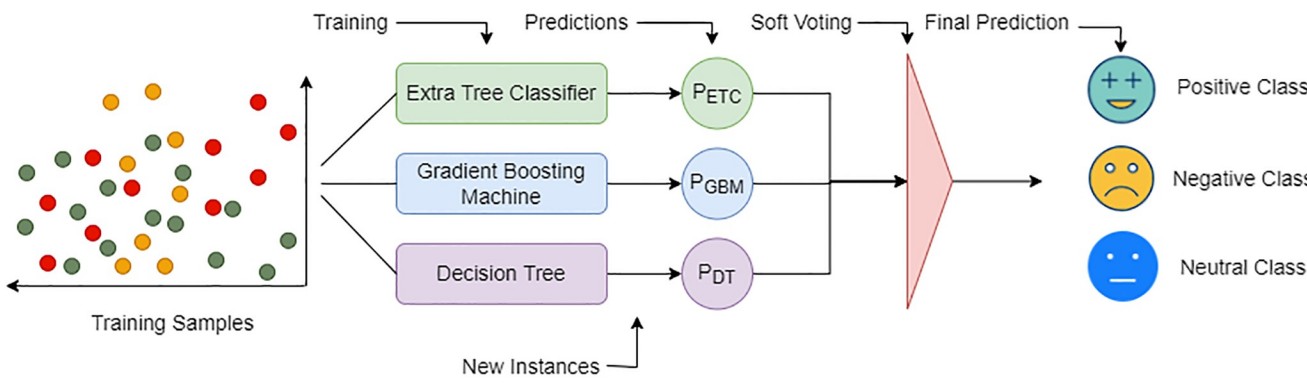

**Fig 4. Architecture of proposed EB-DT voting classifier.**

## Proposed voting classifier

EB-DT is a probabilistic voting classifier that involves an ensemble of three best performing individual classifiers including ETC, GBM, and DT, as shown in Fig 4. The proposed model is based on the combination of three classifiers and carries out prediction tasks using the highest probability of the output variable from each of the three classifiers. The proposed EB-DT combines the performance of individual classifiers into a strong learner [51]. It works by aggregating the output of each of the three classifiers using voting. Afterward, prediction is performed based on the majority voting criterion. EB-DT integrates soft voting to predict the output class by averaging the probability assigned to the classes by each of the individual classifiers.

The choice of an estimated ensemble model stands upon several grounds. First, ensemble models has the tendency to show better performance as reported in the existing literature [32, 45, 55]. Utilizing more than one model potentially reduces the misclassification rate. Secondly, the choice of models varies concerning the nature of data and models can be influenced by the size of the feature set, the ratio of class samples (balanced vs imbalanced datasets), etc. which can increase the probability of model overfit or underfit. Using multiple models on the datasets and combining their output mitigates such risks. Thirdly, the voting criterion is selected because of the results reported for voting classifiers where the ensemble models tend to show better performance than individual models [33, 56]. Keeping in view the diversity of the text tokens used in the tweets, it seems a probable choice to utilize multiple classifiers. Different types of machine learning models like tree-based or linear models, show a different level of performance with sentiment analysis tasks, and using more than one classifier to make the final prediction tends to increase the probability of correct class prediction.

Each model in the proposed EB-DT ensemble takes the text input features to predict probabilities for positive, negative, and neutral sentiments. Mathematically it can be written as

$$C_p = \frac{\sum_{i=1}^{N} avgC_i}{N} \tag{1}$$

where $C_p$ is the prediction probability by the EB-DT using soft voting criteria, $N$ is the number of models, and $avgC$ is the per sentiment probability by each model.

The $C_p$ can also be written as

$$C_p = \frac{C_{DT} + C_{GBM} + C_{ETC}}{N} \tag{2}$$

$$Pos_p = \frac{Pos_{DT} + Pos_{GBM} + Pos_{ETC}}{3} \tag{3}$$

For EB-DT, Eq 3 is used to find the probability of positive sentiment against a tweet. Here $Pos_{DT}$ is the probability of positive class by the DT model, $Pos_{GBM}$ is the probability by the GBM, and $Pos_{ETC}$ is the probability of positive class by the ETC model. Similarly, EB-DT calculates the probability for negative and neutral sentiments as follows

$$Neg_p = \frac{Neg_{DT} + Neg_{GBM} + Neg_{ETC}}{3} \tag{4}$$

$$Neu_p = \frac{Neu_{DT} + Neu_{GBM} + Neu_{ETC}}{3} \tag{5}$$

The ensemble model EB-DT makes the final prediction using

$$EB - DT_p = argmax\{Pos_p, Neg_p, Neu_p\} \tag{6}$$

The functioning of the EB-DT can be further elaborated using an example run. For this purpose, the tweet "One of two shots so far, but have always, and will continue to wear the mask" is taken. DT probabilities against this tweet are (Pos = 0.3, Neg = 0.2, Neu = 0.5), ETC shows probabilities as (Pos = 0.2, Neg = 0.2, Neu = 0.6), while GBM shows (Pos = 0.2, Neg = 0.1, Neu = 0.7) for the given tweet. Using Eq 2, average probabilities can be calculated as

$$Pos_p = \frac{0.3 + 0.2 + 0.2}{3} = 0.23 \tag{7}$$

$$Neg_p = \frac{0.2 + 0.2 + 0.1}{3} = 0.16 \tag{8}$$

$$Neu_p = \frac{0.5 + 0.6 + 0.7}{3} = 0.6 \tag{9}$$

The average probabilities are then used by the argmax function to predict the final class of the tweet which in this case is neutral as it has the highest average probability considering the models that form the ensemble.

$$EB - DT_p = argmax\{0.23, 0.16, 0.6\} \tag{10}$$

$$EB - DT_p = 0.6(Neutral) \tag{11}$$

## Latent Dirichlet allocation

LDA is a well-known probabilistic topic modeling method that is utilized for the extraction of topics from the given input data [57]. The word Latent in LDA conveys the meaning of existing information that is yet to be discovered. The fundamental purpose of topic modeling is to extract the hidden topics using the Dirichlet model which exploits the patterns of similar

words occurring frequently and repeating together. LDA estimates the occurrence of a word in a document which results in building data points and estimating the probabilities. Furthermore, probabilities for the topics in the document are generated which are further classified into the corpus or set of documents [58]. LDA considers two main metrics: 1) The corpus is composed of topics, and 2) The topics are composed of words.

## Results and discussion

The study focuses on analyzing the sentiment and significant topics of tweets related to the COVID-19 safety precautions. Standard comparison of the sentiment classifiers with TF-IDF and BoW features along with the performance of deep learning classifiers is demonstrated. Experiments are conducted using NLTK libraries in Python language. The data is divided into train and test in the ratio of 0.8 to 0.2, respectively.

### Experimental results of machine learning sentiment classifiers

Sentiment classification of tweets is conducted on the preprocessed dataset using TfidfVectorizer(). Table 4 presents the results of machine learning classifiers when trained using the TF-IDF features. Results show that the boosting classifier including GBM and tree-based classifiers including ETC and DT outperform other conventional machine learning models with a 0.97 accuracy score. GBM leverages the regularization methods to reduce the over-fitting of training data resulting in more accurate results. SVC and RF perform slightly poorly in classifying the tweets using TF-IDF features with a 0.95 accuracy score, however, RF yields a higher F1 score of 0.95 as compared to SVC which achieves a 0.93 F1 score. The difference in the performance of RF and ETC is due to the ability of ETC to use randomized features for the training of de-correlated trees instead of the optimal ones which control the over-fitting and enhance the performance of the model. LR performs comparatively lower with a 0.93 accuracy score and 0.91 F1 score. In line with this, AC performs poorly as well with a 0.90 accuracy score and 0.88 F1 score. KNN does not work well with high dimensional data and shows performance with the lowest accuracy score as compared to other classifiers in predicting the sentiments of the tweets.

Fig 5 graphically presents the performance comparison of machine learning sentiment classifiers using TF-IDF features. It can be observed that similar to accuracy, DT, ETC, and GBM achieved optimum precision, recall, and F1 scores. The performance of classifiers in terms of sentiment labels, when combined with TF-IDF is displayed in Fig 6. The machine learning sentiment classifiers perform well for the prediction of the neutral class as the number of records corresponding to the neutral class is higher as compared to the other two classes. Whereas, an overall decrease in the performance of the classifiers can be observed in the prediction of negative class due to a smaller number of records corresponding to negative sentiments.

Apart from TF-IDF, we implemented BoW for sentiment classification experiments. Table 5 demonstrates the results of machine learning classifiers when combined with BoW features. The boosting classifier along with tree-based classifiers including ETC and DT show better performance compared to other classifiers. However, an improvement in the performance of other classifiers can be observed when integrated with BoW features as compared to their performance with TF-IDF features. The BoW feature set is larger as compared to TF-IDF due to its extraction of features irrespective of their position and structure in the text which resulted in the improved performance of the classifiers. The accuracy of SVC and RF has been improved from 0.95 with TF-IDF to 0.96 with BoW. However, the performance of KNN does not improve as it does not work well with high-dimensional data.

**Table 4. Experimental results of machine learning sentiment classifiers using TF-IDF features.**

| Model | Accuracy | Class | Precision | Recall | F1 score |
|---|---|---|---|---|---|
| RF | 0.95 | Negative | 0.98 | 0.79 | 0.87 |
| | | Neutral | 0.93 | 1.00 | 0.96 |
| | | Positive | 0.98 | 0.95 | 0.97 |
| | | Macro avg | 0.96 | 0.91 | 0.95 |
| LR | 0.93 | Negative | 0.97 | 0.72 | 0.83 |
| | | Neutral | 0.91 | 0.99 | 0.95 |
| | | Positive | 0.97 | 0.94 | 0.94 |
| | | Macro avg | 0.95 | 0.88 | 0.91 |
| ETC | 0.97 | Negative | 0.96 | 0.91 | 0.94 |
| | | Neutral | 0.97 | 1.00 | 0.98 |
| | | Positive | 0.98 | 0.96 | 0.97 |
| | | Macro avg | 0.97 | 0.96 | 0.96 |
| GBM | 0.97 | Negative | 0.93 | 0.94 | 0.93 |
| | | Neutral | 0.98 | 0.99 | 0.98 |
| | | Positive | 0.99 | 0.96 | 0.97 |
| | | Macro avg | 0.97 | 0.96 | 0.96 |
| KNN | 0.76 | Negative | 0.70 | 0.54 | 0.61 |
| | | Neutral | 0.76 | 0.84 | 0.80 |
| | | Positive | 0.77 | 0.71 | 0.74 |
| | | Macro avg | 0.75 | 0.70 | 0.72 |
| SVC | 0.95 | Negative | 0.95 | 0.79 | 0.86 |
| | | Neutral | 0.93 | 0.99 | 0.96 |
| | | Positive | 0.98 | 0.94 | 0.96 |
| | | Macro avg | 0.95 | 0.91 | 0.93 |
| AC | 0.90 | Negative | 0.92 | 0.74 | 0.82 |
| | | Neutral | 0.86 | 1.00 | 0.93 |
| | | Positive | 0.98 | 0.83 | 0.90 |
| | | Macro avg | 0.92 | 0.86 | 0.88 |
| DT | 0.97 | Negative | 0.93 | 0.92 | 0.93 |
| | | Neutral | 0.99 | 0.99 | 0.99 |
| | | Positive | 0.97 | 0.97 | 0.97 |
| | | Macro avg | 0.96 | 0.96 | 0.96 |

Fig 7 presents the comparison of the overall performance of sentiment classifiers when combined with BoW features. It can be observed that the tree-based models including RF, DT, and ETC perform comparatively better as compared to the other models in terms of precision, recall, and F1 score. Similarly, GBM also yields the overall highest precision, recall, and F1 score. KNN however performs the worst while SVC, AC, and LR also perform comparatively low as compared to tree-based models and boosting models. However, an improvement in their performance can be observed with the increase in the feature set size by BoW as compared to TF-IDF.

Fig 8 shows that the performance related to the neutral class is comparatively better as compared to the negative and positive classes. This is due to the small number of records of the negative and positive classes and a large number of neutral records. A large number of records enables the classifiers to learn the patterns more effectively thus resulting in better performance for that particular class.

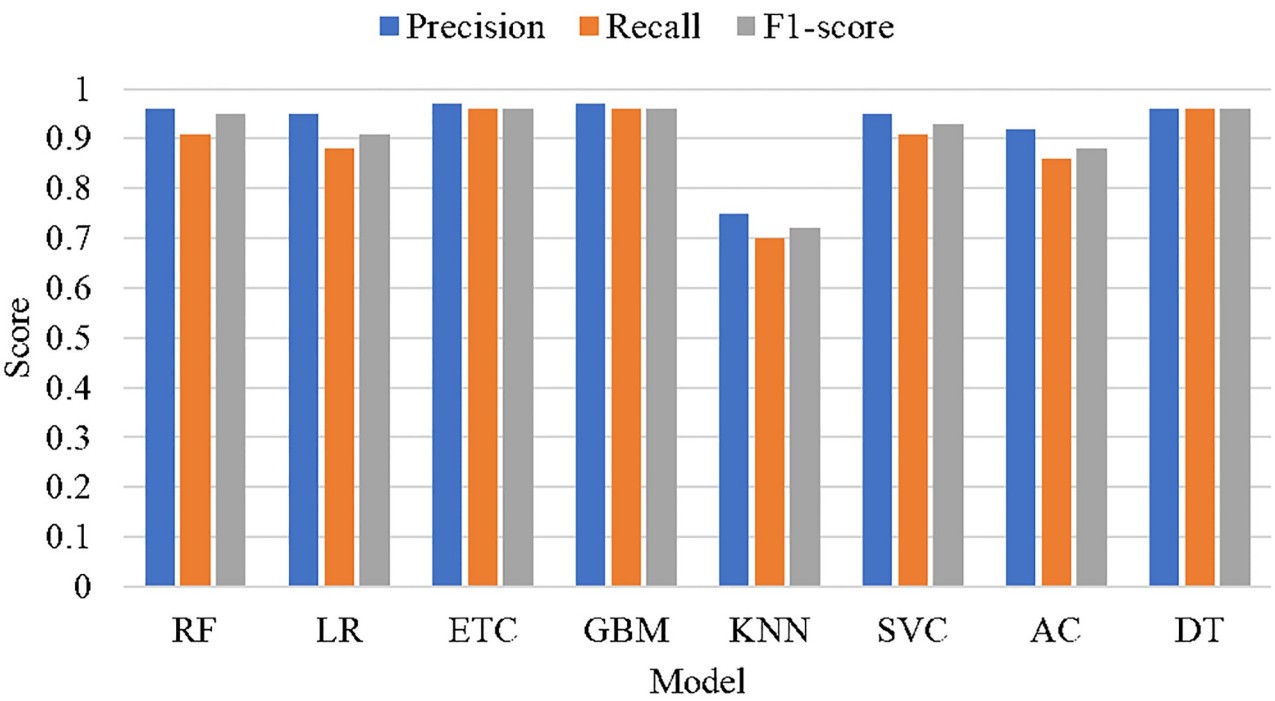

**Fig 5. Experimental results of Sentiment classifiers using TF-IDF features.**

## Experimental results of deep learning classifiers

This study also makes use of three deep learning models including LSTM, GRU, and ensemble model CNN-LSTM. The details of the layer-wise architecture of deep learning models are

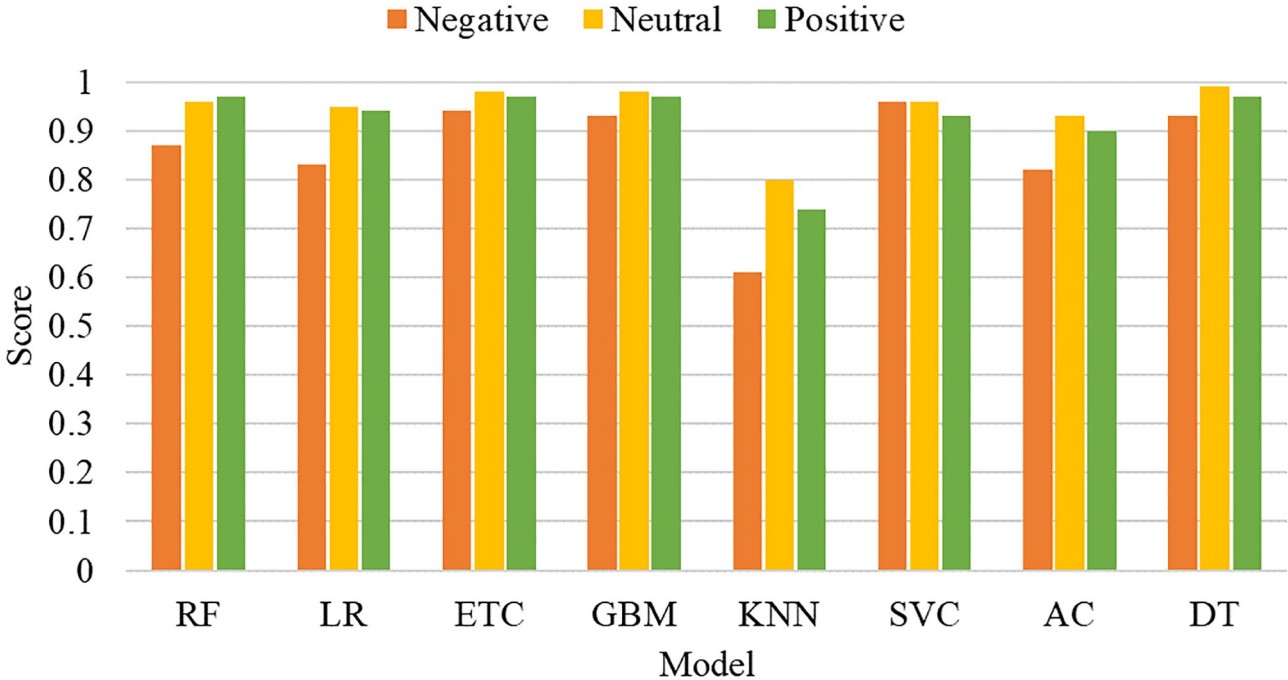

**Fig 6. Performance comparison of classifiers with TF-IDF features for individual class.**

**Table 5. Experimental results of machine learning classifiers using BoW features.**

| Model | Accuracy | Class | Precision | Recall | F1 score |
|---|---|---|---|---|---|
| RF | 0.96 | Negative | 0.98 | 0.86 | 0.91 |
| | | Neutral | 0.95 | 1.00 | 0.97 |
| | | Positive | 0.99 | 0.95 | 0.97 |
| | | Macro avg | 0.97 | 0.94 | 0.95 |
| LR | 0.95 | Negative | 0.95 | 0.80 | 0.87 |
| | | Neutral | 0.94 | 0.99 | 0.96 |
| | | Positive | 0.98 | 0.95 | 0.97 |
| | | Macro avg | 0.95 | 0.92 | 0.93 |
| ETC | 0.97 | Negative | 0.96 | 0.92 | 0.94 |
| | | Neutral | 0.97 | 1.00 | 0.98 |
| | | Positive | 0.98 | 0.97 | 0.97 |
| | | Macro avg | 0.97 | 0.96 | 0.97 |
| GBM | 0.97 | Negative | 0.91 | 0.90 | 0.90 |
| | | Neutral | 0.97 | 0.99 | 0.98 |
| | | Positive | 0.98 | 0.95 | 0.96 |
| | | Macro avg | 0.97 | 0.97 | 0.97 |
| KNN | 0.71 | Negative | 0.87 | 0.37 | 0.52 |
| | | Neutral | 0.65 | 0.98 | 0.78 |
| | | Positive | 0.95 | 0.44 | 0.61 |
| | | Macro avg | 0.82 | 0.60 | 0.64 |
| SVC | 0.96 | Negative | 0.93 | 0.86 | 0.89 |
| | | Neutral | 0.96 | 0.99 | 0.97 |
| | | Positive | 0.98 | 0.96 | 0.97 |
| | | Macro avg | 0.96 | 0.94 | 0.95 |
| AC | 0.92 | Negative | 0.92 | 0.76 | 0.83 |
| | | Neutral | 0.90 | 0.99 | 0.95 |
| | | Positive | 0.96 | 0.88 | 0.92 |
| | | Macro avg | 0.93 | 0.88 | 0.90 |
| DT | 0.97 | Negative | 0.98 | 0.93 | 0.95 |
| | | Neutral | 0.99 | 1.00 | 0.99 |
| | | Positive | 0.98 | 0.98 | 0.98 |
| | | Macro avg | 0.98 | 0.97 | 0.98 |

provided in Fig 9. All models use an embedding layer at the beginning with an input dimension of 100 and a vocabulary size of 5000. LSTM uses a dropout layer with a 0.5 drop rate, followed by an LSTM layer, and a dense layer. GRU also uses the same dropout layer, followed by a GRU layer, a simple RNN (Recurrent Neural Network) layer, and a dense layer in the end. The combined CNN-LSTM network uses a Conv1D layer after the embedding layer which is followed by a max-pooling layer. In the end, it uses two dense layers of 16 and 3 neurons, respectively. All models use the 'categorical_crossentropy' loss function while the 'Adam' optimizer is used.

Experimental results for deep learning models given in Table 6 show that LSTM shows an accuracy of 0.97 as compared to GRU and CNN-LSTM each with an accuracy score of 0.96. With its ability to learn longer dependencies LSTM outperforms GRU and CNN. As this study is concerned with the multi-class classification problem, we will also consider the F1 score of the classifiers for a better understanding of their performance. It can be observed that the

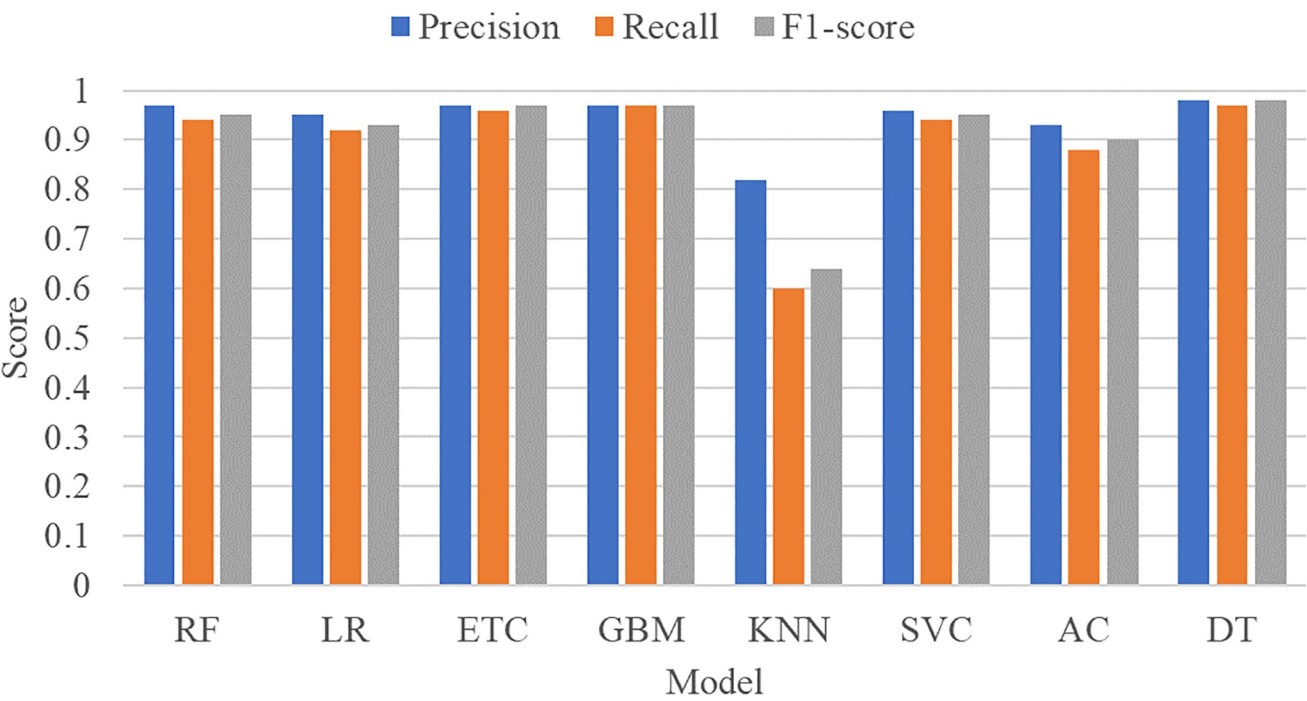

**Fig 7. Experimental results of machine learning classifiers using BoW features.**

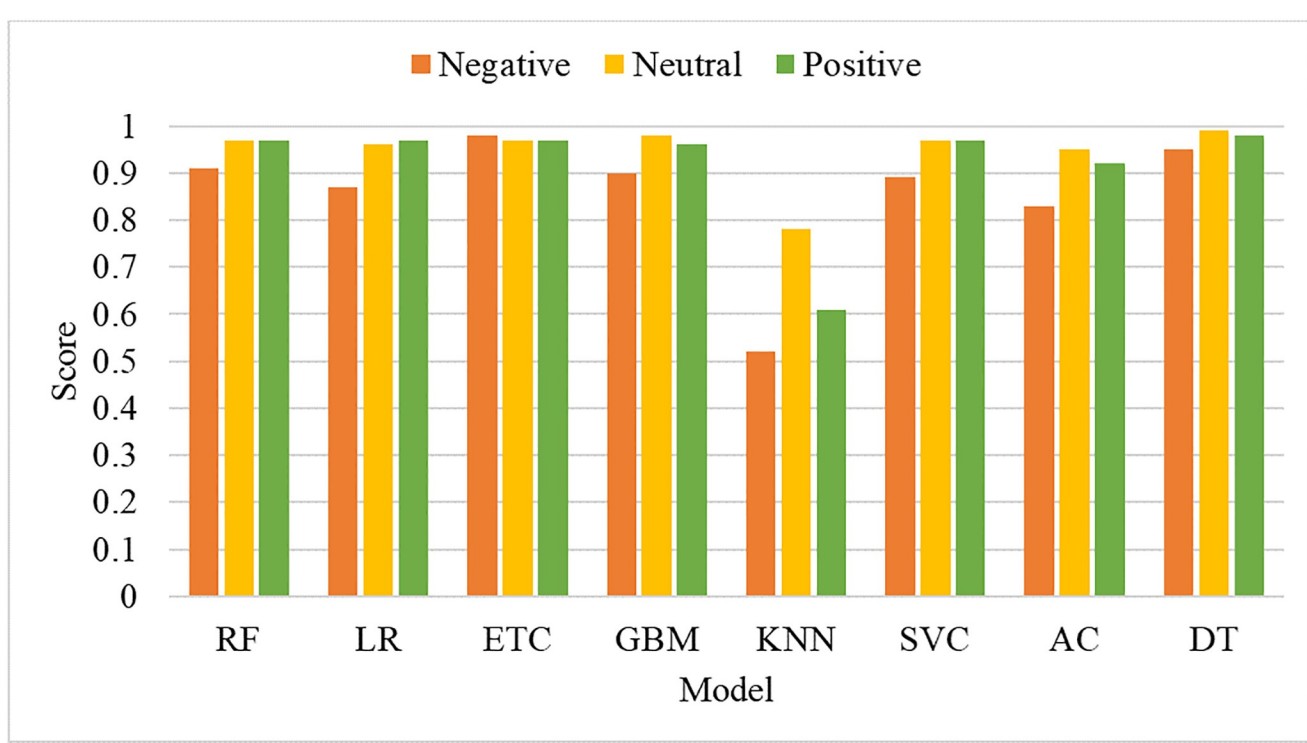

**Fig 8. Performance analysis for individual class using BoW features.**

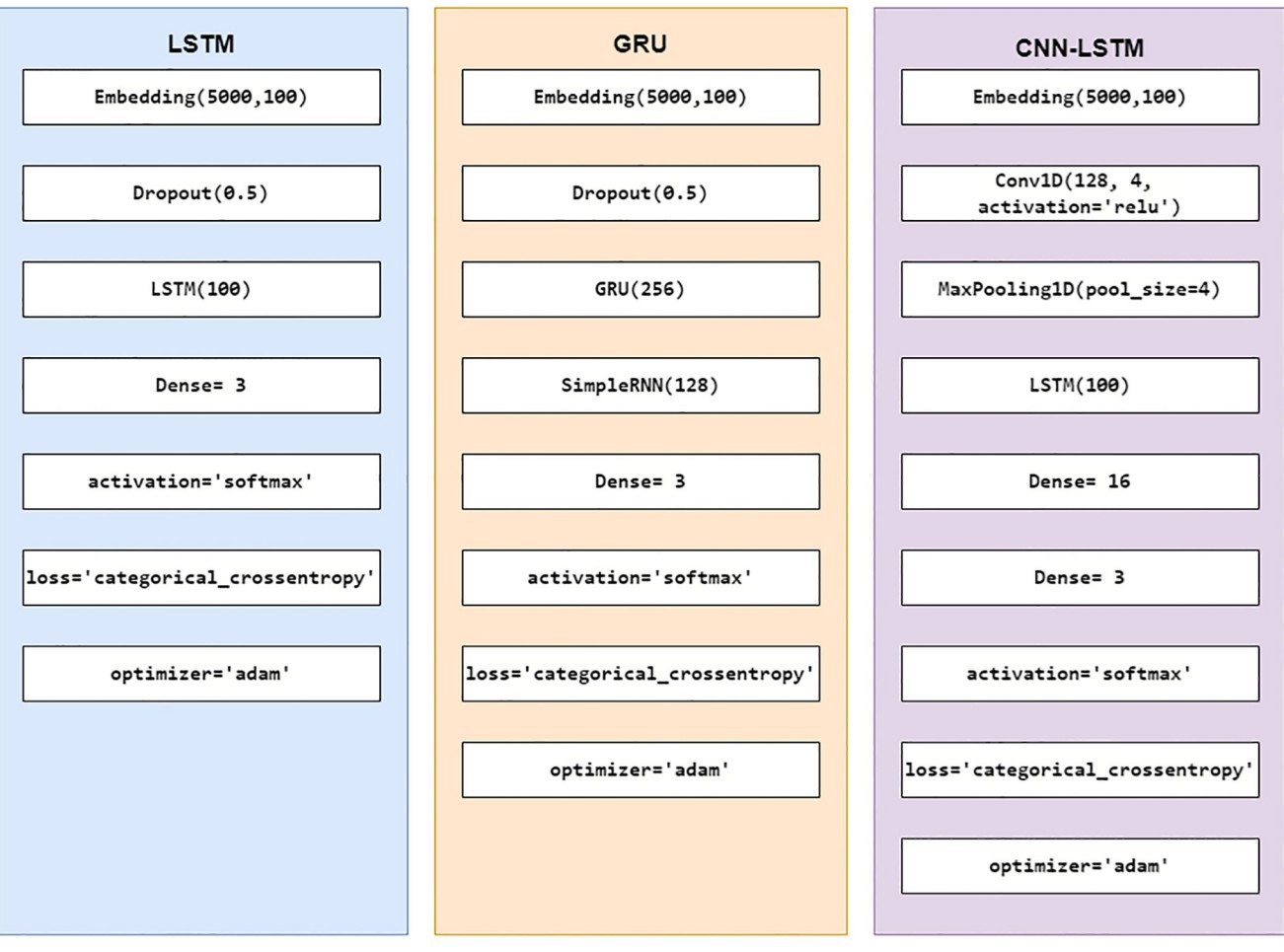

**Fig 9. Layered architecture of deep learning models.**

**Table 6. Experimental results of deep learning classifiers in sentiment classification of tweets.**

| Model | Accuracy | Class | Precision | Recall | F1 score |
|---|---|---|---|---|---|
| LSTM | 0.97 | Negative | 0.95 | 0.88 | 0.91 |
|  |  | Neutral | 0.98 | 0.98 | 0.98 |
|  |  | Positive | 0.96 | 0.98 | 0.97 |
|  |  | Macro avg | 0.96 | 0.95 | 0.95 |
| GRU | 0.96 | Negative | 0.91 | 0.94 | 0.93 |
|  |  | Neutral | 0.98 | 0.96 | 0.97 |
|  |  | Positive | 0.96 | 0.97 | 0.96 |
|  |  | Macro avg | 0.95 | 0.96 | 0.95 |
| CNN-LSTM | 0.96 | Negative | 0.94 | 0.90 | 0.92 |
|  |  | Neutral | 0.98 | 0.96 | 0.97 |
|  |  | Positive | 0.94 | 0.98 | 0.96 |
|  |  | Macro avg | 0.95 | 0.95 | 0.95 |

macro average F1 score of three of the deep learning models is 0.95. However, the classifiers show better performance in the prediction of the neutral class. The deep learning models perform well when subjected to a large dataset. The underlying dataset in this study consists of 8911 records which maybe not be large enough to obtain the highest performance as compared to the tree-based RF, DT, and ETC classifiers. CNN and GRU fail to outperform the traditional machine learning classifiers due to their low learning and slow convergence.

## Experimental results of proposed EB-DT classifier

The tree-based models including DT, and ETC showed robust performance along with the boosting classifier GBM as discussed previously. Therefore, this study combines the best performing classifiers to further improve the overall performance of the resulting ensemble. For this purpose, we proposed a voting classifier EB-DT which is an ensemble of three base classifiers including ETC, DT, and GBM. The resulting EB-DT classifier makes final predictions by aggregating the predictions of base classifiers under the soft voting criteria.

Table 7 shows the experimental results of the proposed EB-DT using TF-IDF and BoW features. It can be observed that the proposed EB-DT classifier shows robust performance with a 0.99 accuracy, along with similar macro average precision, recall, and F score. However, the performance of the proposed EB-DT model decreased when combined with the BoW feature set. TF-IDF extracts features that have high predictive information regarding the target variable. However, the feature extraction technique BoW extracts all of the known vocabulary words from the data irrespective of their relevance in the whole corpus which results in lower performance of the proposed EB-DT. In the case of sentiment labels, it can be observed that the F1 score for the neutral class is 0.99 with both TF-IDF and BoW feature sets while the negative class, performs comparatively lower with a 0.96 F1 score for both features. Results confirm that using the voting method to combine the outputs of three highly performing classifiers yields the best result regarding the accuracy, precision, recall, and F1 score and achieves the highest accuracy of 0.99 which is higher than both the traditional machine learning and deep learning classifiers.

## Performance analysis of sentiment classifiers

Fig 10 shows the accuracy score of all the sentiment classifiers utilized for experiments in this study. It can be seen that the proposed model EB-DT shows highly accurate results in comparison with other classifiers when combined with TF-IDF. It also performs well with BoW features as compared to other classifiers when used with BoW features. Furthermore, deep learning models perform comparatively lower than the proposed voting classifier. EB-DT leverages the benefits of DT, ETC, and GBM and achieves highly accurate results for classifying

**Table 7. Experimental results of the proposed EB-DT with TF-IDF and BoW features.**

| Features | Accuracy | Class | Precision | Recall | F1 score |
|---|---|---|---|---|---|
| TF-IDF | 0.99 | Negative | 0.99 | 0.94 | 0.96 |
| | | Neutral | 0.98 | 1.00 | 0.99 |
| | | Positive | 0.99 | 0.98 | 0.98 |
| | | Macro avg | 0.99 | 0.99 | 0.99 |
| BoW | 0.98 | Negative | 0.99 | 0.94 | 0.96 |
| | | Neutral | 0.98 | 1.00 | 0.99 |
| | | Positive | 0.99 | 0.98 | 0.98 |
| | | Macro avg | 0.99 | 0.97 | 0.98 |

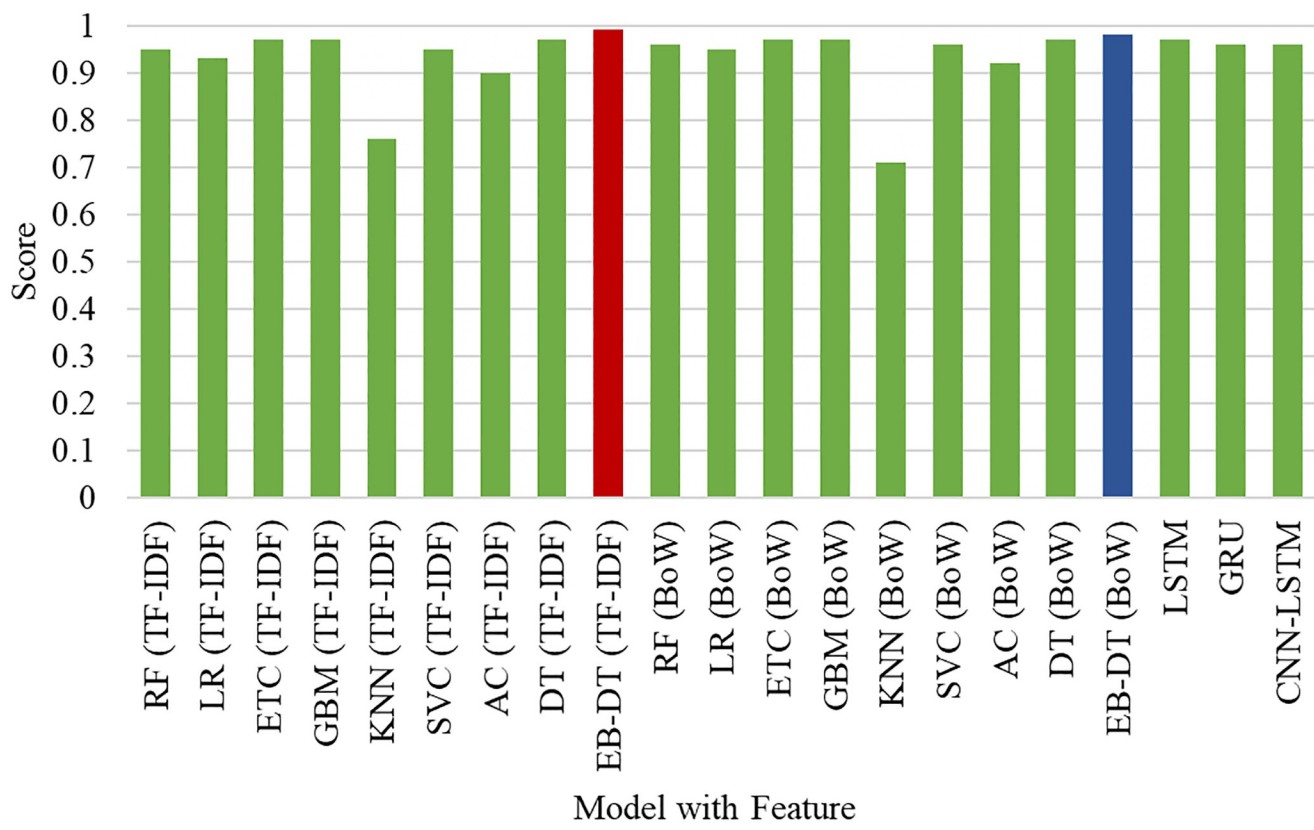

**Fig 10. Performance comparison of classifiers in sentiment classification of tweets.**

the sentiments of extracted tweets. Overall, the performance of KNN is observed to be the lowest as it does not work well with high dimensionality data. As the text features extracted by TF-IDF and BoW have a high dimension which decreases the performance of KNN. Whereas, the boosting classifier AC also does not perform for both TF-IDF and BoW features.

### Proposed approach comparison with previous studies

To show the significance of the proposed approach, the performance comparison is carried out with existing state-of-the-art studies. Several studies are selected for comparisons, such as [32] proposed an ensemble model SVE for the medical review sentiment predictions. The study [33] used voting criteria to combine the GBM and SVM for user reviewer sentiment classification. Similarly, the study [34] used soft voting criteria to combine LR and SGDC models. Performance comparison results are presented in Table 8 indicating the superior performance of the

**Table 8. Performance comparison with previous studies.**

| Reference | Model | Accuracy | Precision | Recall | F1 score | Execution time |
|---|---|---|---|---|---|---|
| [32] | SVE | 0.97 | 0.97 | 0.95 | 0.96 | 17.6 Sec |
| [33] | GBSVM | 0.96 | 0.95 | 0.95 | 0.95 | 16.2 Sec |
| [34] | LR-SGDC | 0.93 | 0.95 | 0.91 | 0.92 | 6 Sec |
| Current study | EB-DT | 0.99 | 0.99 | 0.99 | 0.99 | 16.1 Sec |

**Table 9. Topics extracted from positive and negative tweets using LDA.**

| Topics | Negative Keywords | | | | Positive Keywords | | | |
|---|---|---|---|---|---|---|---|---|
| 1. | mouth | cover | headach | adjust | best | love | cute | good |
| 2. | nose | fuck | unbreathable | dark | safe | clean | love | cover |
| 3. | comfortable | work | face | double | reusable | love | clean | safe |
| 4. | mouth | cloth | cover | face | Safe | like | cover | best |
| 5. | unbreathable | dark | lightweight | difficult | wear | love | best | fit |

proposed approach with a 0.99 accuracy score which is far better than existing works. In addition, the robustness of the proposed approach is evaluated by comparing its execution time.

The proposed model is better in terms of computational time as compared to other studies. Although [34] provides a much smaller execution time, its accuracy is substantially low as compared to the proposed model. Also, the current study uses three models while existing studies use two models for the ensemble.

## Topic modelling

The positive and negative tweets are subjected to LDA to extract topics that correspond to the factors impacting the sentiments of the public opinions towards following COVID-19 safety precautions. We extracted a total of 5 topics from both positive and negative tweets individually, as shown in Table 9. A total of 40 most occurring words are also extracted from the tweet dataset using LDA. The resulting keywords for positive sentiments are "love, cute, good, mouth, nose, comfortable, clean, best, fit, cover" and the main keywords for negative sentiments are "fuck, work, dark, nose, cover, adjust, unbreathable". From the extracted topic keywords, it can be judged that the negative sentiments appeared due to the masks being unbreathable causing discomfort in wearing them. Another cause for the negative sentiments can be the double-layered facemasks covering the mouth and nose of people which causes difficulty in speaking and breathing. However, from the keywords extracted from positive sentiments, it can be observed that people felt safe by wearing the masks and they find it fascinating as they breathe clean air. The re-usability of the masks is also one of the main factors causing the positive opinions from the public.

Fig 11 shows the percentage ratio of positive and negative key words found using LDA. It can be observed that 'resuabl' and 'wear' have the highest ratio for positive terms while 'bad' and 'fuck' are the most frequently used words among the found negative terms.

## Analyzing change in sentiments over Time-I

This study further analyzes peoples' sentiments regarding COVID-19 restrictions over time to study the impact of time on the change in tweets' sentiment. For this purpose, a separate dataset is collected after one month for the analysis. The same data extraction procedure is adopted as before and tweets are extracted for 17 February when the new recorded COVID-19 cases were low. We analyze the data using the proposed sentiment analysis approach and a third-party application 'sentiment viz' [59]. Fig 12 shows the sentiment count on the new data. It can be seen that an increasing trend is observed for positive tweets. Apparently, the people realize the effectiveness and benefit of wearing a facemask against COVID-19 which shifts their sentiments in favor of these restrictions. Similarly increase in the negative sentiments has been observed, however, the ratio of positive sentiments is substantially higher than both neutral and negative tweets.

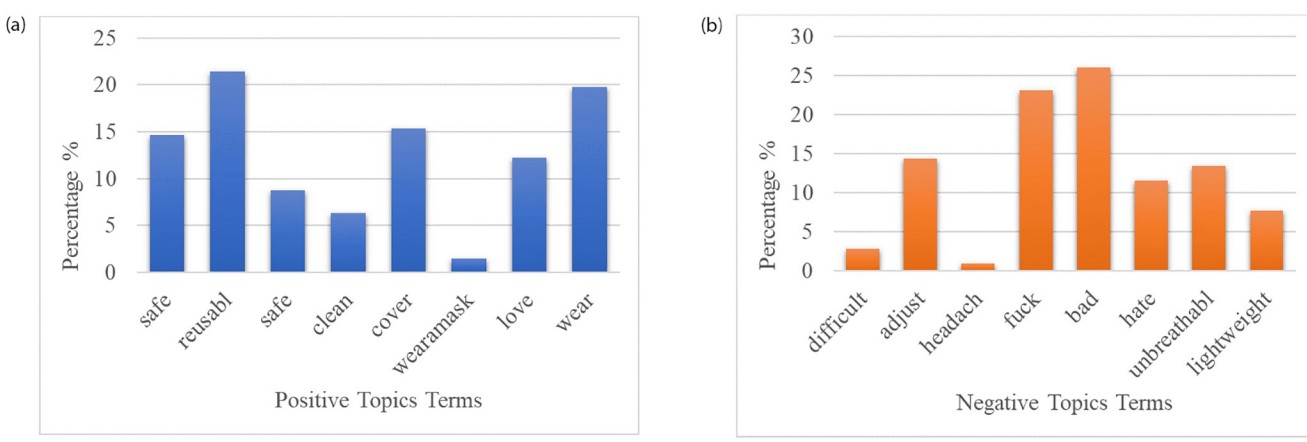

**Fig 11. Positive and negative tweets topics term percentage in data, (a) Positive Topics Terms, (b) Negative Topics Terms.**

## Analyzing change in sentiments over Time-II

The third set of experiments is performed for analyzing the change in COVID-19 restrictions-related sentiments on the data collected for 19 February 2022. Comparative results of positive, negative, and neutral sentiments are shown in Fig 13. Trends for peoples' sentiments show that a gradual increase in positive sentiments is prevalent. Over time, peoples' sentiments have shifted from neutral to negative and positive. However, tweets containing positive sentiments are higher than negative sentiments.

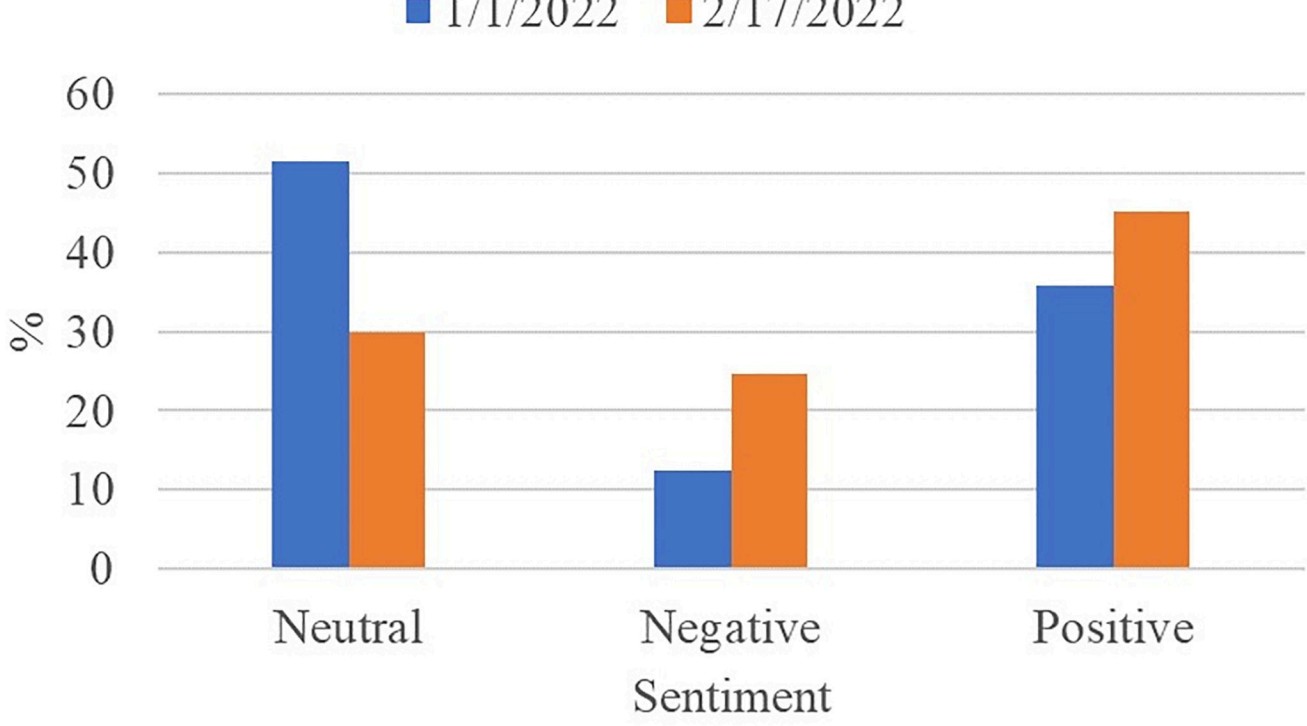

**Fig 12. Comparison of positive, negative and neutral tweets on 1st January and 17 February 2022.**

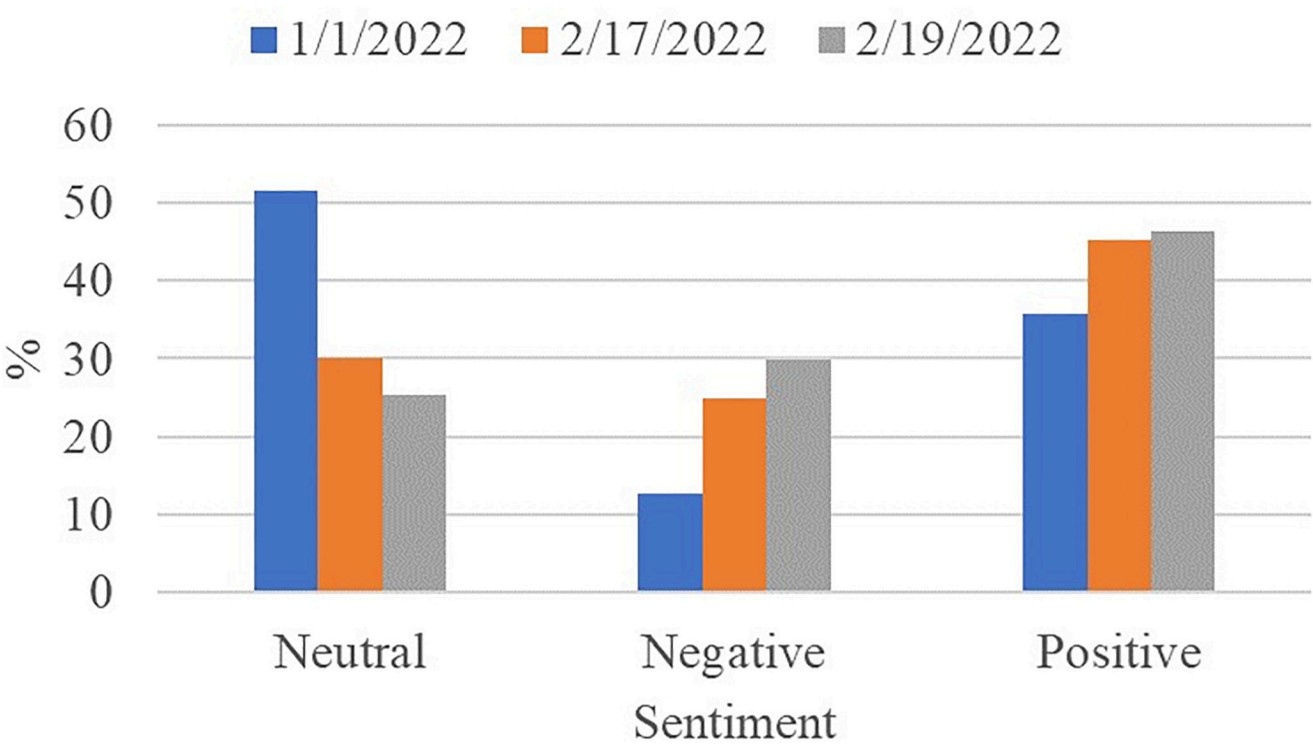

**Fig 13. Comparison of positive, negative and neutral tweets on 1st January, 17 February and 19 February 2022.**

## Comparison of proposed approach results with third-party solutions

For corroborating the results of the current study, a separate analysis is performed using a third-party solution, called 'sentiment viz'. Figs 14–16 show the sentiment viz app results on the new dataset. A high number of happy emotions are observed in the new dataset. Results from sentiment viz confirm the results obtained using the proposed approach. Figs 14 and 15 show that tweets show calm, relaxed and happy emotions for COVID-19 precautions as compared to before.

Fig 16 shows the word cloud for the new dataset. It shows that the pleasant category contains the safety, beneficiary, etc. words while the unpleasant category contains annoying, risk, death, etc. words. These words also confirm the words found using topic modeling with LDA.

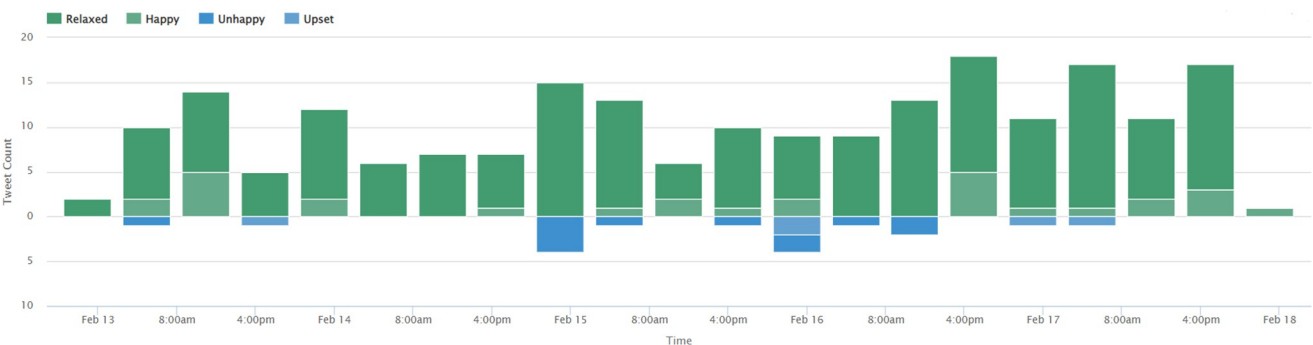

**Fig 14. People emotions in new tweets dataset extracted using sentiment viz app.**

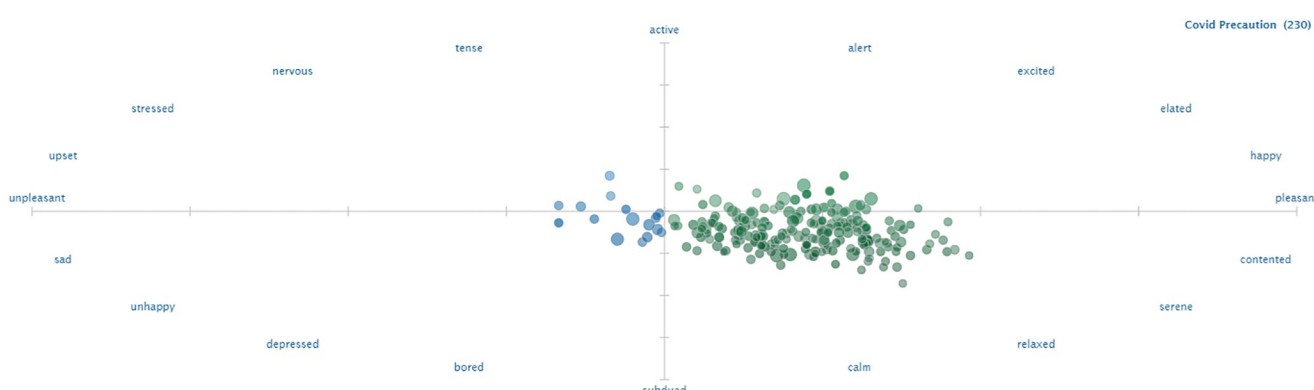

**Fig 15. People emotions and sentiments in new tweets dataset extracted using sentiment viz app.**

## Statistical T-test

We also deploy a T-test to show the statistical significance of the proposed EB-DT voting classifier [60]. Statistical T-test compares the proposed EB-DT results with other state-of-the-art models. In the output, if the critical value is greater than or equal to the t-stat then T-test accepts the null hypothesis ($H_o$) indicating that the T-test rejects the alternative hypothesis which shows that both compared results are not statistically significant.

When the results of EB-DT are compared with other models' results using TF-IDF the T-test gives a t-stat value of 7.483 and a critical value of 0 which shows that the T-test rejects the $H_o$. It indicates that the results from the proposed EB-DT model are statistically significant. When we compared the EB-DT results with other models' results using the BoW feature the T-test gives a t-stat value of 3.667 and a critical value of 0 which also shows that the T-test rejects the $H_o$ to show the statistical significance of the proposed approach.

## Policy implications

This study is designed to analyze the reviews of the public regarding COVID-19 restrictions. Although such restrictions serve as the first line of defense and are initiated for the safety of the people from the pandemic, people do not share the same sentiment. Consequently, people show their willingness and dissent on social media platforms. Analyzing such views can provide useful insights into people's sentiments and provide an overall picture of what people think of these restrictions. Consequently, current policies about the COVID-19 restrictions

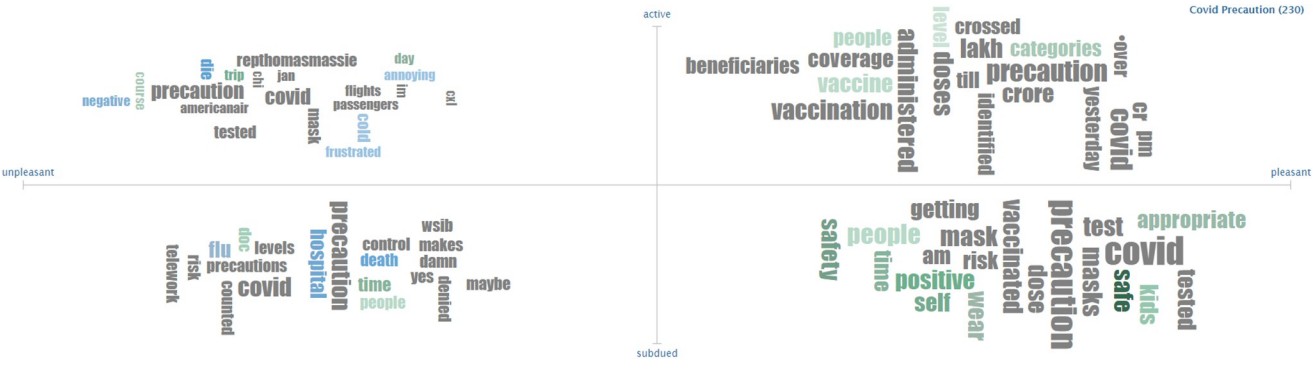

**Fig 16. Word cloud in terms of pleasant and unpleasant categories.**

can be drafted and revised accordingly. Additionally, the analysis of change in people's sentiment over time can be beneficial to examine the impact of specific policies for a particular period. The study does not define any such policy or analyze it, it simply provides insight into people's sentiments. Since people's views and sentiments are deemed important, especially in developed countries, they do have an impact on the decision-making process and can be used to take corresponding actions.

## Conclusion

The rapid COVID-19 outbreak inflicted large human and financial loss with the increasing number of infections that further continues to grow large with each passing day. The policy shifts by the government have resulted in the polarization of the underlying topic of the COVID-19 precautionary measures. The limited size of data in the case of surveys confines the scope of the study and a large dataset is required to analyze the trends over time. For this, this study gathered multiple datasets over different periods from Twitter containing tweets related to COVID-19 safety precautions. For analysis, an automated hybrid approach is proposed to investigate the sentiments of the public towards COVID-19 safety restrictions. Extensive experiments are performed to analyze the performance of machine learning using TF-IDF and BoW and deep learning models in comparison to the proposed EB-DT model. Experimental results show that the overall performance of machine learning classifiers is better with BoW features. The proposed EB-DT outperforms both machine learning and deep learning models with a 0.99 accuracy score when used with TF-IDF features. Moreover, this study uses topic modeling and 5 topics each for positive and negative tweets are extracted. Regarding the negative sentiments, with the mouth and nose covered by the facemask, people feel difficulty in breathing and condemn the restriction of wearing the facemask. On the other hand, positive tweets commend the use of facemask to be safe from viruses, and filters and provide clean air to breathe. Analyzing the change in tweets sentiments over time reveals that peoples' sentiments regarding COVID-19 restrictions have shifted from neutral to negative and positive. However, the tweets containing positive sentiments are substantially higher than tweets containing negative tweets.

This study incorporates a dataset of a total of 8911 extracted tweets that may not be enough to analyze the performance of the deep learning algorithms as they tend to work well with large datasets. In addition, not all the extracted tweets are reliable as the number of fake accounts has drastically increased over time on social media platforms. This might introduce bias in the sentiments of the extracted tweets. Similarly, despite being the leading social media platform, Twitter does not contain all the views regarding COVID-19, and considering the views and comments from other social media platforms like Facebook, Reddit, etc. may change the results.

A potential future work in this study is the utilization of part-of-speech tagging to remove the words which are not relevant to the sentiments such as nouns that do not contribute to the sentiment of the text. Another possible future work can be the identification of the fake accounts to reduce the bias in the dataset. The sentiment classification of social media texts enables governments to gain insights into the public opinion towards policy changes thus allowing them to make informed decisions. The results in this study are limited to the data related to Twitter users. Therefore, we plan on incorporating public opinions from a variety of social media platforms to provide generalizability to the proposed model.

## Author Contributions

**Conceptualization:** Ayaz Ahmad, Furqan Rustam.

**Data curation:** Ayaz Ahmad, Eysha Saad.

**Formal analysis:** Furqan Rustam, Eysha Saad, Muhammad Abubakar Siddique, Imran Ashraf.

**Funding acquisition:** Isabel de la Torre Díez.

**Investigation:** Muhammad Abubakar Siddique.

**Methodology:** Ayaz Ahmad, Muhammad Abubakar Siddique.

**Project administration:** Arturo Ortega Mansilla.

**Resources:** Isabel de la Torre Díez.

**Software:** Arturo Ortega Mansilla.

**Supervision:** Imran Ashraf.

**Validation:** Ernesto Lee, Isabel de la Torre Díez.

**Visualization:** Ernesto Lee, Arturo Ortega Mansilla.

**Writing – original draft:** Furqan Rustam, Eysha Saad.

**Writing – review & editing:** Imran Ashraf.

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
