## [Decision Letter · Decision Letter 0]

2 May 2022

PONE-D-22-09735Analyzing Preventive Precautions to Limit Spread of COVID-19PLOS ONE

Dear Dr. Imran,

Thank you for submitting your manuscript to PLOS ONE. After careful consideration, we feel that it has merit but does not fully meet PLOS ONE’s publication criteria as it currently stands. Therefore, we invite you to submit a revised version of the manuscript that addresses the points raised during the review process.

ACADEMIC EDITOR: Authors are requested to address the comments of the reviewer. Furthermore, the manuscript should be refined for English grammatical structure and phraseology. The manuscript should be polished by an English linguist or language service. 'Grammarly' is poor quality and not acceptable.

We look forward to receiving your revised manuscript.

Kind regards,

Usman Qamar

Academic Editor

PLOS ONE

Journal Requirements:

2. Please remove the username and location of the twitter accounts in Table 1." 2) "Please note that PLOS ONE has specific guidelines on code sharing for submissions in which author-generated code underpins the findings in the manuscript. In these cases, all author-generated code must be made available without restrictions upon publication of the work. Please review our guidelines at https://journals.plos.org/plosone/s/materials-and-software-sharing#loc-sharing-code and ensure that your code is shared in a way that follows best practice and facilitates reproducibility and reuse.

"“This research was supported by the European University of The Atlantic. This research was supported by the Florida Center for Advanced Analytics and Data Science funded by Ernesto.Net (under the Algorithms for Good Grant)."

"This research was supported by the European University of The Atlantic. This research 586

was supported by the Florida Center for Advanced Analytics and Data Science funded 587

by Ernesto.Net (under the Algorithms for Good Grant)."

"This research was supported by the European University of The Atlantic. This research was supported by the Florida Center for Advanced Analytics and Data Science funded by Ernesto.Net (under the Algorithms for Good Grant)."

5.We note that you have indicated that data from this study are available upon request. PLOS only allows data to be available upon request if there are legal or ethical restrictions on sharing data publicly. For more information on unacceptable data access restrictions, please see http://journals.plos.org/plosone/s/data-availability#loc-unacceptable-data-access-restrictions. 

Reviewers' comments:

Reviewer's Responses to Questions

**Comments to the Author**

1. Is the manuscript technically sound, and do the data support the conclusions?

Reviewer #1: No

2. Has the statistical analysis been performed appropriately and rigorously? 

Reviewer #1: No

3. Have the authors made all data underlying the findings in their manuscript fully available?

Reviewer #1: No

4. Is the manuscript presented in an intelligible fashion and written in standard English?

Reviewer #1: No

5. Review Comments to the Author

Reviewer #1: The Introduction should highlight the relevance of the topic, the novelty of the results, the importance of policy implications, the sample’s choice, the methodology’s appropriateness, the data used, the contribution to the literature, and the limitations of the study.

The literature review is partial and incomplete, and some recent and relevant contributions should be cited and discussed: i.e., 10.1017/S095026882100248X; 10.1016/j.jenvman.2021.112241; 10.1007/s11356-020-10689-0; 10.1016/j.envres.2020.110663; 10.1016/j.apenergy.2020.115835.

The theoretical framework should be discussed more in detail.

The estimated model must be justified in light of the literature on this specific topic.

Descriptive statistics are absent.

Diagnostic tests are absent.

Robustness checks are absent.

The results should be discussed more in detail.

Comparisons with previous studies are absent.

Conclusions are too short.

Policy implications are weak.

Further research should be indicated.

Limitations of the study are not provided.

The English needs a proofreading by a native speaker.

The editing does not follow the journal’s guidelines.

The originality value of the study is limited.

This is an applied exercise without a clear innovative intuition.

6. PLOS authors have the option to publish the peer review history of their article (what does this mean?). If published, this will include your full peer review and any attached files.

Reviewer #1: No

---

## [Author Response · Author response to Decision Letter 0]

4 Jul 2022

A separate file has been added containing response to reviewers' comments.

---

## [Decision Letter · Decision Letter 1]

19 Jul 2022

Analyzing Preventive Precautions to Limit Spread of COVID-19

PONE-D-22-09735R1

Dear Dr. Ashraf,

We’re pleased to inform you that your manuscript has been judged scientifically suitable for publication and will be formally accepted for publication once it meets all outstanding technical requirements.

Kind regards,

Usman Qamar

Academic Editor

PLOS ONE

Additional Editor Comments (optional):

Reviewers' comments:

Reviewer's Responses to Questions

**Comments to the Author**

1. If the authors have adequately addressed your comments raised in a previous round of review and you feel that this manuscript is now acceptable for publication, you may indicate that here to bypass the “Comments to the Author” section, enter your conflict of interest statement in the “Confidential to Editor” section, and submit your "Accept" recommendation.

Reviewer #1: All comments have been addressed

2. Is the manuscript technically sound, and do the data support the conclusions?

Reviewer #1: Yes

3. Has the statistical analysis been performed appropriately and rigorously? 

Reviewer #1: Yes

4. Have the authors made all data underlying the findings in their manuscript fully available?

Reviewer #1: Yes

5. Is the manuscript presented in an intelligible fashion and written in standard English?

Reviewer #1: Yes

6. Review Comments to the Author

Reviewer #1: ---------------------------------------------------------------------------------------------------------------------------------------------

7. PLOS authors have the option to publish the peer review history of their article (what does this mean?). If published, this will include your full peer review and any attached files.

Reviewer #1: No

---

## [Editor Report · Acceptance letter]

4 Aug 2022

PONE-D-22-09735R1 

Analyzing Preventive Precautions to Limit Spread of COVID-19 

Dear Dr. Ashraf:

I'm pleased to inform you that your manuscript has been deemed suitable for publication in PLOS ONE. Congratulations! Your manuscript is now with our production department. 

Kind regards, 

on behalf of

Dr. Usman Qamar 

Academic Editor

PLOS ONE